# Fecal Metagenomics and Metabolomics Identifying Microbial Signatures in Non-Alcoholic Fatty Liver Disease

**DOI:** 10.3390/ijms24054855

**Published:** 2023-03-02

**Authors:** Satu Pekkala

**Affiliations:** 1Faculty of Sport and Health Sciences, University of Jyväskylä, 40014 Jyväskylä, Finland; satu.p.pekkala@jyu.fi; Tel.: +358-453582898; 2Institute of Biomedicine, Medical Microbiology and Immunology, University of Turku, 20014 Turku, Finland

**Keywords:** gut microbiota, metabolomics, metagenomics, liver fat, NAFLD, diet, metabolic pathways

## Abstract

The frequency of non-alcoholic fatty liver disease (NAFLD) has intensified, creating diagnostic challenges and increasing the need for reliable non-invasive diagnostic tools. Due to the importance of the gut–liver axis in the progression of NAFLD, studies attempt to reveal microbial signatures in NAFLD, evaluate them as diagnostic biomarkers, and to predict disease progression. The gut microbiome affects human physiology by processing the ingested food into bioactive metabolites. These molecules can penetrate the portal vein and the liver to promote or prevent hepatic fat accumulation. Here, the findings of human fecal metagenomic and metabolomic studies relating to NAFLD are reviewed. The studies present mostly distinct, and even contradictory, findings regarding microbial metabolites and functional genes in NAFLD. The most abundantly reproducing microbial biomarkers include increased lipopolysaccharides and peptidoglycan biosynthesis, enhanced degradation of lysine, increased levels of branched chain amino acids, as well as altered lipid and carbohydrate metabolism. Among other causes, the discrepancies between the studies may be related to the obesity status of the patients and the severity of NAFLD. In none of the studies, except for one, was diet considered, although it is an important factor driving gut microbiota metabolism. Future studies should consider diet in these analyses.

## 1. Introduction

WHO estimates that over 1.9 billion adults worldwide are overweight, and consequently, the frequency of metabolic disorders, including non-alcoholic fatty liver disease (NAFLD) has exacerbated. Metabolic disorders include a cluster of physiological conditions, including increased blood pressure, high blood sugar, excess body fat around the waist, and abnormal cholesterol or triglyceride levels, which often occur together. Alarmingly, these diseases now also extend to children, in addition to the young and middle-aged population [1]. In Western countries, up to 90% of the obese population is estimated to suffer from NAFLD [2]. In Nordic countries, NAFLD is the second-most increasing indication for liver transplantation, and thus, it is a great burden to the healthcare system. Therefore, new diagnostic tools allowing early detection of the disease would be of great importance.

NAFLD is defined as excessive fat accumulation (over 5% fat in hepatocytes) in the liver, without secondary causes of fat accumulation, such as the excessive drinking of alcohol and treatment with steatogenic drugs (e.g., methotrexate). Histopathologically, NAFLD can be categorized into simple steatosis (non-alcoholic fatty liver, NAFL), which is diagnosed as a presence of hepatic fat accumulation without any histological or biochemical injuries, and non-alcoholic steatohepatitis (NASH), which is characterized by steatosis, inflammation, and hepatocyte damage, i.e., ballooning, and can be accompanied by cirrhosis or not [3,4]. It is estimated that 3 to 5% of NAFLD patients can develop NASH [5].

The high prevalence of NAFLD creates diagnostic challenges, and there is a growing need for reliable non-invasive diagnostic tools. Due to the importance of the gut–liver axis in the onset and progression of NAFLD [6], several recent studies have attempted to reveal the microbial signatures in NAFLD, to evaluate their suitability as diagnostic biomarkers of NAFLD, and to predict the progression of the disease. This article reviews the evidence of microbial signatures in NAFLD that have been analyzed using fecal metagenomics and metabolomics. Metagenomics refers to the analysis of gut microbiota composition and functional genes using shotgun sequencing. Metabolomics refers to quantification of the fecal metabolites using either nuclear magnetic resonance (^1^H-NMR) or ultra-high performance liquid chromatography (LC)/mass spectrophotometry (MS). Ultimately, the pitfalls and caveats of such approaches will be summarized, as well as the avenues for future directions.

The literature searches for this review article were made between September and December 2022. The search words “fecal AND metabolomics AND (liver fat OR NAFLD OR NASH)” and “fecal AND metagenomics AND (liver fat OR NAFLD OR NASH)” were used in both PubMed and Ovid Medline. Animal studies were omitted from the search results.

## 2. Non-Invasive, Cost-Effective, and Easy Diagnostics of NAFLD for Clinical Settings

Due to the fact that liver biopsies are highly invasive, the development of new cost-effective diagnostic tools for NAFLD is important. What makes the early diagnosis difficult is that before the onset of severe fibrosis or cirrhosis, the NAFLD patients may remain asymptomatic. Therefore, the diagnosis is often made coincidentally due to abnormal findings in routine blood samples. Values above the upper limit of normal serum alanine aminotransferase (ALT, ~40 IU/L in men and ~30 IU/L in women), as well as abnormally high serum triglycerides and LDL cholesterol, can be an indication of NAFLD [7]. However, according to some studies, the liver enzymes may be completely normal in most patients [8].

At present, there are several available non-invasive methods to diagnose NAFLD; however, these do not enable the distinguishing of steatosis from steatohepatitis, nor do they evaluate the severity of hepatic fibrosis. In this context, several panel markers, indexes, and scores have been developed for diagnostics. For instance, the liver fat score includes as measured variables, including the presence of metabolic syndrome and type 2 diabetes (T2D), fasting serum insulin, serum aspartate aminotransferase (AST), and the AST/ALT ratio [9]. The fatty liver index (FLI) considers body mass index (BMI), waist circumference, serum triglyceride levels, and gamma-glutamyltransferase (GGT) in the general population, with low prevalence of T2D [10]. To distinguish NASH from NAFL, the HAIR score, which includes the determination of the presence of hypertension, elevated ALT, and insulin resistance, has been employed [11]. An advanced fibrosis scoring system, developed and validated by McPherson et al., is generated by determining age, hyperglycemia, BMI, platelet count, serum albumin, and the AST/ALT ratio [12].

However, measuring variables of blood and body composition is insufficient for the ultimate diagnosis, and thus, imaging techniques and other analyses are needed. While ultrasonography is the most cost-effective and suitable method in clinical practice, it may fail to detect mild steatosis. One study showed that ultrasound was unable to detect steatosis present in less than 10% of hepatocytes [13]. In addition, the visual assessment of NAFLD by ultrasonography exhibits significant substantial inter-observer variability, which limits the reproducibility of the results [14].

## 3. Physiological and Molecular Players in the Onset of NAFLD—Future for Diagnostics

It is now well accepted that the pathogenesis of NAFLD involves multiple simultaneous “hits” associated with environmental, host genetic, and physiologic factors [15], as opposed to the initially proposed “two-hit theory” [16]. The multiple hit hypothesis implies that simple hepatic steatosis may be a benign process, and NASH might be a separate disease with a different pathogenesis. The pathogenic hits include: (1) inflammatory mediators derived from various tissues [17,18]; (2) increased lipid storage, lipogenesis, and (adipo)cytokines that activate endoplasmic reticulum stress [19,20,21]; (3) mitochondrial dysfunction [21] and reactive oxygen species due to lipotoxicity [22,23]; (4) nutrient sensing [24]; and (5) genetic factors [25,26,27,28]. Importantly, all these events may occur together rather than consecutively. Recent studies have also highlighted the importance of connective tissue dysfunction and insulin resistance in the onset of NAFLD [29,30,31]. However, if these above-described factors were used for diagnostic purposes, they would all require invasive sampling; thus, non-invasive alternatives are of interest.

Nearly a decade ago, we [32] and Mouzaki et al. [33] were pioneers in showing that gut microbiota composition associates with hepatic fat content in humans. There are recent excellent reviews on the topic [34]; therefore, this review will concentrate on the advances in fecal metabolomics and metagenomics related to NAFLD. It is increasingly accepted that the abundance and functions of many members of the gut microbiota affect human physiology by processing the ingested food into certain bioactive metabolites [35]. These molecules can act as inter-tissue signaling messengers by penetrating the portal vein and subsequently, the liver, to promote or prevent hepatic fat accumulation (Figure 1). Collecting fecal samples for gut microbiota composition and microbial metabolite analyses would be an excellent non-invasive method for NAFLD diagnostics, which will be reviewed in the upcoming sections. While this review concentrates on the human gut microbiota metabolism (fecal metabolomics and metagenomics) and not animal studies, one important host molecular mechanism that connects the gut microbiota and its metabolism to NAFLD is briefly introduced below as an example of animal models.

One mechanistic animal study showed that gut microbiota-dependent hepatic lipogenesis was mediated by hepatic stearoyl CoA desaturase 1 (SCD1) [36]. The authors used germ-free and conventional Toll-like receptor 5 (TLR5) deficient (T5KO) mice, which are prone to develop microbiota-dependent metabolic syndrome, to first show that the T5KO mice displayed elevated hepatic neutral lipid content, depending on the presence of gut microbiota. TLR5 recognizes flagellin, which is the structural protein of the locomotive organelle of bacteria [37]. After the initial observations, Singh et al. found that colonic short-chain fatty acids (SCFA) receptors, as well as hepatic lipogenic enzymes, including SCD1, were upregulated in T5KO mice, and that gut-derived SCFA were increasingly incorporated into palmitate in the liver. Dietary SCFA further aggravated hepatic steatosis and metabolic syndrome, which were impeded by the hepatic deletion of SCD1. All the above-mentioned effects were ablated in the germ-free mice, but when the germ-free mice were transplanted with the cecal microbiota of T5KO mice, their hepatic palmitate content doubled. The authors concluded that while several beneficial properties have been recognized for SCFA, their excess in conditions combined with innate immune deficiency and dysbiotic, overgrown microbiota due to T5KO may increase susceptibility to metabolic diseases [36].

## 4. Fecal Metabolomics and Metagenomics Identifying the Microbial Signatures in NAFLD

The biological system effects of gut microbial metabolism are excellently reviewed elsewhere [35], and therefore, in this review the emphasis will be on NAFLD. It is likely that the severity of NAFLD affects the functions of the gut microbiota differently, and vice versa. Therefore, the reviewed studies are divided below into subsections by the disease severity (steatotic, without diagnosed NAFLD, NAFL, NASH, fibrosis, cirrhosis, or hepatocellular carcinoma). Ultimately, studies in children with NAFLD are also reviewed. The reviewed studies are listed in Table 1, which is primarily divided according to the sections below, and by the order of appearance of the references in the text.

### 4.1. Fecal Metabolomics and Metagenomics Identifying the Gut Microbial Signatures in Steatotic Adults without Diagnosed NAFLD

By far, the largest metagenomic study conducted in fatty liver disease included a sample of 6269 Finnish participants, most of which were overweight, but according to the BMI, there were normal weight individuals included as well [38]. It should be noted that the functional metagenomic profiling was qualitative and not quantitative because the sequencing depth did not allow for the assembling of contigs. In addition, instead of measuring liver fat content with imaging, the authors used FLI to categorize the participants into liver fat content groups. Nevertheless, FLI is a rather widely used and accepted index for NAFLD and its stratification [39]. Ethanol and the SCFA acetate production pathways were found to be positively associate with FLI [38]. Previously, it has been shown, for instance, that high-alcohol-producing strains of *Klebsiella pneumoniae* exist in humans with NAFLD [40]. Thus, it seems that endogenously produced alcohol may play a role in hepatic fat accumulation, at least in some populations. Generally, SCFA are considered to exert beneficial functions in the host, such as modulating immune functions [41] and gastrointestinal permeability [42]. However, they may also negatively impact the inflammatory status of the host [36].

Fecal metagenomic signatures have been described in non-diabetic morbidly obese women with hepatic steatosis [43]. By mapping the microbial genes into the Kyoto Encyclopedia of Genes and Genomes (KEGG) modules, it was found that steatosis associated positively with carbohydrate, lipid, and amino acid metabolism, as well as lipopolysaccharide (LPS) and peptidoglycan biosynthesis. LPS are recognized by Toll-like receptor 4 (TLR4), which expression has been shown to be increased in the livers of obese patients with NASH [44]. Peptidoglycan, in turn, is recognized by multiple pattern-recognition receptors, including nucleotide-binding oligomerization domain-containing proteins (NODs), domain-containing 3 (NLRP3), and Toll-like receptor 2 (TLR2) [45]. In addition to the above-mentioned pathways, Hoyles et al. also observed that hepatic steatosis was associated with an increased number of genes related to the biosynthesis of branched-chain amino acids (BCAA) and aromatic amino acids [43]. The metagenomic findings of feces were further supported by the elevated concentrations of these particular amino acids in plasma. To mechanistically show that the microbiome contributes to hepatic steatosis, the authors transplanted feces from steatotic human donors into mice. Indeed, the fecal transplants caused rapid hepatic fat accumulation in mice, which involved elevated circulating BCAA [43]. These observations are interesting, as numerous studies have linked BCAA to obesity and NAFLD (for review, see [46]).

We recently compared fecal and plasma metabolomes of humans with low (<5% of fat in liver) and high (>5% of fat in liver) liver fat content, without diagnosed NAFLD [47]. The study groups did not differ from each other in BMI. We found that the fecal histidine metabolism product, N-omega-acetylhistamine, was markedly increased in individuals with fatty liver disease. In addition, another product of histidine degradation, anserine, positively associated with liver fat content [47]. Previously, plasma levels of histidine have been shown to associate with the grade of hepatic steatosis [48]. In agreement with a previous metagenome study [49], we found that the levels of lysine degradation product, saccharopine, were higher in the feces of individuals with high liver fat content [47]. As an indication of decreased steroid metabolism, the fecal levels of 6-hydroxybetatestosterone were reduced in the steatotic individuals. In contrast, a previous study reported that low serum testosterone levels were associated with hepatic steatosis in obese males [50].

In summary, the two metagenomic studies in individuals with hepatic steatosis show different microbial signatures, which may be due to the differences in obesity status between the study populations. Neither of the metagenomic studies determined dietary factors between the groups or used them as confounding factors [38,43]. In our metabolomic study, there were no major dietary differences between the high and low liver fat content groups, except that higher vitamin E and sucrose intake was observed in individuals with fatty livers [47]. Interestingly, sucrose is known to contribute to the onset of NAFLD [51], while vitamin E is considered as a possible treatment for NAFLD [52]. The main findings of the three reviewed studies in this section are presented in Figure 2.

### 4.2. Fecal Metabolomics and Metagenomics Identifying the Gut Microbial Signatures in Patients with NAFL, NASH, and Hepatic Fibrosis

Ge et al. studied fecal metabolomics in patients with NAFLD and healthy controls using UHPLC/MS/MS [53]. All participants were overweight and/or had visceral obesity. Combining the metabolite identification with KEGG pathway analysis, the authors found that the metabolism of nicotinate, nicotinamide, and pyrimidine, as well as signaling pathways of calcium and oxytocin and pancreatic secretion were altered in NAFLD patients compared to healthy controls. Interestingly, nicotinamide adenine dinucleotide (NAD^+^) is being studied in clinical trials as a potential target to treat NAFLD [54]. However, Ge et al. failed to show differences in the fecal levels of nicotinate between the healthy and NAFLD groups [53]. Altered pyrimidine metabolism was reflected in lower fecal levels of uracil in the NAFLD patients. In addition, the catabolic byproduct of purine metabolism, xanthine, was also lower, likely due to the lower enzymatic activity of xanthine oxidase. The authors concluded that xanthine, along with the abundance of specific microbial taxa, might contribute to the diagnostics of NAFLD [53]. Intriguingly, it has been shown that the inhibition of xanthine oxidase can ameliorate hepatic steatosis in mice [55]. This might be related to the decreased production of reactive oxygen species (ROS) [56].

Based on the literature, it seems that microbial functional genes can differentiate less advanced NAFLD from NASH and NAFLD according to the presence of significant fibrosis [57] and the stage of fibrosis [58]. However, the authors of one paper did not perform shotgun metagenome sequencing but predicted the functional potential of the gut microbiota from 16S rRNA gene data using PICRUSt [57]. The name is an abbreviation for Phylogenetic Investigation of Communities by Reconstruction of Unobserved States. PICRUSt is a bioinformatics software package designed to predict metagenome functional content from marker gene (e.g., 16S rRNA gene) surveys and full genomes using an algorithm developed by Languille et al. [59]. However, the enriched functional categories in NASH were mostly related to carbohydrate and lipid metabolism [57], which is similar to what has been found by others in simple steatosis [43] and in NASH [60]. Interestingly, patients with fibrosis could be distinguished from the patients without fibrosis by their microbial functional genes. Fibrosis was associated with enriched functional categories related to carbohydrate and lipid metabolism [57]. A study that combined real microbial metagenomics and the analysis of plasma metabolites in obese NAFLD patients [58] did not find similar microbial signatures in fibrosis, as did PICRUSt [57]. Of the set of 89 metabolites that are produced by both the host and the microbiota, 11 metabolites could differentiate between mild/moderate NAFLD and advanced fibrosis. These enriched genes/metabolites were mainly related to carbon metabolism in fibrosis grade 2, as well as to nucleotide and steroid degradation in fibrosis grade 1. In addition, higher SCFA butyrate was annotated in grade 1, while the SCFA butyrate and propionate were higher in grade 2. Many findings from plasma metabolites and fecal metagenomic pathways supported each other [58]. Contrary to these findings, using fecal metabolomics, we have found that testosterone metabolism in the gut is lower in obese individuals with fatty liver disease [47].

In addition to the stage of hepatic fibrosis, the obesity status of the patients appears to be linked to the distinct gut microbial features in NAFLD [61]. Lee et al. reported that while not seen in obese individuals with NAFLD, the levels of SCFA acetate and propionate increased along with the stage of fibrosis in non-obese individuals [61]. This again indicates that the SCFA are not necessarily always beneficial for the host’s health. Lee et al. further found that the fecal levels of several conjugated and unconjugated bile acids were higher in non-obese individuals with fibrosis, while in obese individuals, the levels of total conjugated bile acids were inversely associated with the severity of fibrosis [61]. Supporting the findings of fecal metabolomics, the expression levels of microbial genes encoding for bile salt hydrolase and 7α-hydroxy-3-oxochol-4-en-24-oyl-CoA dehydrogenase were lower in the non-obese patients with fibrosis.

The contribution of the bile acids to the pathophysiology of NAFLD via the gut–liver axis is rather well established, yet the bile acids have been mostly analyzed in plasma and not in feces in NAFLD studies [62]. However, contrary to the findings of Lee et al. [61] reviewed above, Smirnova et al. reported that the fecal concentrations of several secondary bile acids were lower in obese NASH patients with advanced fibrosis [63]. Yet, there is a difference between the two studies in the obesity status of the patients. In the study by Smirnova et al., taurine conjugated bile acids in particular increased along with the stage of fibrosis [63]. Interestingly, when the authors looked at the NAFLD activity scores (NAS), the findings were the opposite. Compared to the healthy controls, the NAFLD patients had higher fecal levels of secondary bile acids and expression of microbial genes involved in the biotransformation of bile acids. Further, NASH patients had higher levels of conjugated bile acids than patients with NAFL [63]. This, in turn, is contrary to another study by Sui et al., which showed higher levels of primary bile acids, chenodeoxycholic acid, and cholic acid in the feces of non-diabetic individuals with NASH compared to healthy controls [64]. Nevertheless, these patients were of normal weight, and therefore, the obesity status may be related to the different results between the reviewed studies. Of the other fecal metabolites, increased kynurenine and decreased L-tryptophan levels were good predictors of hepatic steatosis [64]. The latter finding is interesting as a previous study had shown that tryptophan-derived metabolite kynurenine causes hepatic steatosis in mice by activating aryl hydrocarbon receptor signaling [65].

In summary, while fecal bile acids seem to play an important role in NAFLD and fibrosis, the reviewed studies present different findings, which may, at least partly, depend on the obesity status of the study populations. None of the studies reported whether there were differences in dietary intakes between the study groups, although one study [64] collected food consumption data. The main findings of the microbial metagenomes and fecal metabolomics of the studies reviewed in this section are presented in Figure 3.

### 4.3. Fecal Metabolomics and Metagenomics Identifying the Gut Microbial Signatures in Patients with NAFLD-Cirrhosis and NAFLD-Hepatocellular Carcinoma

With the global rise in the incidence of obesity and type 2 diabetes, NAFLD-related hepatocellular carcinoma (HCC) and NAFLD-cirrhosis are also becoming common liver diseases [66]. By using metagenomics, Behary et al. showed that the gut microbiome of the NAFLD-HCC patients was characterized by a higher number of SCFA synthesizing genes [67], which is in line with what was found related to an earlier stage of the disease, namely higher FLI [38]. Compared to the NAFLD-cirrhosis patients and healthy controls, genes related to acetate (phosphate acetyltransferase, *pta*), butyrate (phosphate butyryltransferase, *ptb*), and propionate (fumarate reductase, *frd*, and succinate-CoA synthetase, *scs*) synthesis were over-expressed in the feces of the NAFLD-HCC patients [67]. The gene expression findings were further confirmed using targeted LC/MS/MS and ^1^H-NMR quantification of the fecal metabolites. The levels of acetate and butyrate, as well as oxaloacetate and acetylphosphate, which are intermediates of the SCFA metabolism, were higher in the feces of the NAFLD-HCC patients, while the propionate concentration did not differ between the study groups [67]. In the future, it would be interesting to investigate which other metabolites characterize the NAFLD-HCC patients using a non-targeted metabolomic profiling.

In NAFLD-cirrhosis, the gut microbial signatures have been studied by Oh et al. [68]. Interestingly, similar to what has been shown in steatosis without diagnosed NAFLD [43], BCAA and aromatic amino acids were predictors of cirrhosis, as shown by both metagenomic and metabolomic analyses [68]. In the fecal samples of NAFLD-cirrhotic patients, the levels of L-tryptophan were increased due to its decreased metabolism. This is contrary to another study showing that decreased L-tryptophan levels were good predictors of hepatic steatosis [64].

To summarize the literature reviewed above, in NAFLD-associated cirrhosis and HCC, some metabolic signatures similar to those in hepatic steatosis without diagnosed NAFLD can be found, namely higher fecal levels of SCFA, BCAA, and aromatic amino acids. Thus, in the future it would be important to further explore their role and to consider whether these fecal metabolites would be suitable early biomarkers of advanced liver diseases before its onset. It should be noted, however, that as in the other studies reviewed above, none of the studies in this section considered dietary differences as confounding factors between the study groups.

### 4.4. Fecal Metagenomics and Metabolomics Identifying the Gut Microbial Signatures in Children with NAFLD

A multi-omics study by Michail et al. combined metagenomic, metabolomic, and proteomic approaches to study the gut microbiome of obese children, with and without NAFLD [49]. In contrast to one targeted metabolomics study showing higher fecal acetate and propionate levels in NAFLD patients [69], the ^1^H-NMR quantification of the fecal metabolites revealed that acetate levels were lower in pediatric NAFLD patients, while propionate was unaffected [49]. As reviewed above, again supporting the role of endogenously produced ethanol in NAFLD, the fecal levels of ethanol were ~2-fold higher in obese children with NAFLD than in the healthy children and obese children without NAFLD. Based on the microbial metagenomic analysis, the lysine degradation pathway was exclusively identified in children with NAFLD and not in healthy individuals [49]. This agrees with our metabolomic findings from adults, showing higher degradation products of lysine in the feces of individuals with high liver fat content [47]. The metagenome shotgun sequencing of Michail et al. suggested that the microbial energy metabolism, including fatty acid and carbohydrate biosynthesis, is much more enhanced in NAFLD pediatric patients compared to healthy children [49]. However, this might be solely due to the different obesity status of the patients in the studies, as it has been shown that an obese microbiome more efficiently harvests energy from the diet [70].

Testerman et al. studied the fecal metagenomes of children with and without NAFLD [71]. They observed that multiple pathways for lysine synthesis were increased, and histidine degradation was decreased in NAFLD patients. This is contrary to our recent metabolomics findings in adults with fatty liver, showing higher fecal levels of lysine and histidine degradation products [47]. However, in the NAFLD patient’s metagenomes, microbial metabolic pathways for BCAA, aromatic amino acids, and peptidoglycan synthesis were also enriched [71]. Interestingly, similar findings have been reported in morbidly obese adults with hepatic steatosis [43]. Thus, it could be that these metabolic pathways might help in the detection of the disease at an early age and/or stage of the disease.

Another metagenomic study compared healthy children, obese children, and obese children with NAFLD [72]. Compared to healthy children, the obese pediatric patients with and without NAFLD had a lower abundance of microbial genes related to the pathways of replication and reparation, folding, sorting, and metabolism of amino acids. The presence of NAFLD differentiated the obese groups, showing the enrichment of pathways related to the digestive system, immune system, and glycan biosynthesis [72]. Thus, in this case, there were microbial signatures in NAFLD that were likely not dependent on the obesity status of the patients.

In agreement with the findings of Hoyles et al. in adults [43], Kordy et al. report that compared to the healthy BMI-matched individuals, the microbiome of the pediatric patients with NASH was characterized by increased carbohydrate, lipid, and amino acid metabolism, as well as LPS biosynthesis [60]. While the panels of plasma metabolites could accurately predict NASH, this was not seen for the fecal metabolites analyzed from rectal swabs. The author of this current review suggests that the rectal swabs, even if stored at correct temperatures, may not be a reliable way to preserve the fecal metabolites for subsequent analyses.

A combination of fecal metagenomics and targeted UPLC-MS/MS identified alterations in bile acid metabolism in children with NAFLD [73]. The number of genes related to the biosynthesis of secondary bile acids was more abundant in NAFLD patients, which agrees with one study in adults [63]. However, the comparison between the metagenomic and metabolomic results is a bit confusing, as the fecal levels of many secondary bile acids (a-hyodeoxycholic acid, 7-ketolithocholic acid, 23-nordeoxycholic acid, 7,12-diketolithocholic acid, 3-epideoxycholic acid, and dehydrocholic acid) were reduced in NAFLD, while only chenodeoxycholic acid-3-b-D-glucuronide was increased [73]. Unfortunately, the authors did not discuss these discrepant results in their publication.

One study in children with NAFL and NASH used metagenomic shotgun sequencing to reveal microbial signatures related to the disease [74]. Compared to the BMI-matched healthy controls, microbial LPS biosynthesis was significantly enriched in children with NAFL and NASH. This finding agrees with what has been found in non-diabetic obese women with hepatic steatosis [43]. Several genes related to flagellar assembly were also enriched in children with NASH and in those with moderate-to-severe fibrosis [74]. We have previously shown in a mouse model that gut microbial flagellin causes hepatic fat accumulation, which is mediated by vascular adhesion protein-1 [75]. Altogether, these results suggest that flagellin may promote NAFLD, while other studies have suggested that a knockout of flagellin-recognizing TLR5 protects mice from NAFLD [36]. However, it should be noted that flagellin can also act via cytosolic nucleotide oligomerization domain (NOD)-like receptors to affect the inflammatory status [76]. Thus, the effects of flagellin and its receptors in relation to the onset of NAFLD should be further explored in the future.

In summary, studies in children with NAFLD have reported microbial signatures, some of which are similar (increased LPS biosynthesis and amino acid metabolism) and some of which are distinct from each other. However, a few of the reviewed findings are similar to those seen in adults, such as higher fecal levels of ethanol in individuals with hepatic steatosis and increased LPS biosynthesis in NAFLD. None of the reviewed studies in children analyzed whether there were dietary differences between the study groups. The main microbial metagenomic and fecal metabolomic findings from children with NAFLD are summarized in Figure 4.

**Figure 4 ijms-24-04855-f004:**
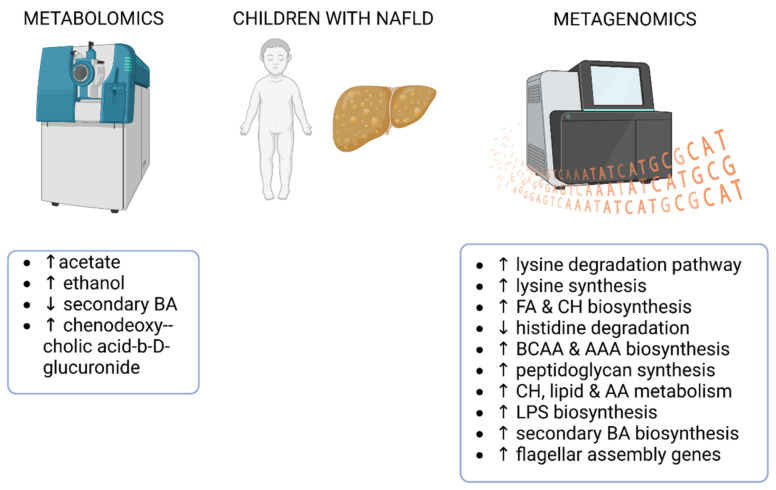
Main findings from studies reporting fecal metagenomics and/or metabolomics in children with NAFLD. FA—fatty acids; CH—carbohydrates; BA—bile acids; AA—amino acids, BCAA—branched chain amino acids; AAA—aromatic amino acids; LPS-lipopolysaccharides. The figure was created using BioRender.

**Table 1 ijms-24-04855-t001:** The reviewed studies presenting metabolomic and/or metagenomic findings regarding the human gut microbiota and NAFLD.

Reference Number in the Text	Analysis Method	Study Patients/Adults
Ruuskanen et al. [38]	metagenomics	fatty liver and all normal weight, overweight, or obese
Hoyles et al. [43]	metagenomics	fatty liver, morbidly obese
Driuchina et al. [47]	metabolomics	fatty liver and healthy, obese
Ge at al. [53]	metabolomics	NAFLD and healthy, overweight
Boursier et al. [57]	predicted metagenomics	NAFLD and NAFLD + fibrosis and NASH
Loomba et al. [58]	metagenomics	NAFLD + fibrosis
Lee et al. [61]	metabolomics	NAFLD + obesity + fibrosis and NAFLD + normal weight + fibrosis
Smirnova et al. [63]	metabolomics	NAFL, NASH, fibrosis, regardless of BMI
Sui et al.	metabolomics	NASH and healthy, normal weight
Behary et al. [67]	metabolomics, metagenomics	NAFLD-HCC and healthy
Oh et al. [68]	metabolomics, metagenomics	NAFLD-cirrhosis and healthy
**Reference number** **in the text**	**Analysis method**	**Study patients/children**
Michail et al. [49]	metabolomics, metagenomics	NAFLD + obese, healthy + obese and healthy + normal weight
Testerman et al. [71]	metagenomics	NAFLD and healthy, obese
Zhao et al. [72]	metagenomics	NAFLD and healthy, obese and normal weight
Kordy et al. [60]	metabolomics, metagenomics	NAFL, NASH and healthy, obese and normal weight
Yu et al. [73]	metabolomics, metagenomics	NAFLD and healthy, normal weight
Schwimmer et al. [74]	metagenomics	NAFL, NASH and healthy, overweight

## 5. Conclusions and Future Directions

Due to the increasing incidence of NAFLD, there is a growing need for non-invasive diagnostic tools. Because of the importance of the gut–liver axis in the pathophysiology of NAFLD, there is the hope that signatures of microbial metabolism could be used for diagnostic purposes. However, studies on the microbial metagenomics and fecal metabolomics in humans with NAFLD are surprisingly scarce. The existing studies reviewed here present mostly distinct, and even contradictory, findings on the microbial metabolites and functional genes in NAFLD. The most abundant reproducing markers reviewed here are increased LPS and peptidoglycan biosynthesis, enhanced degradation of lysine, increased levels of BCAA, as well as altered lipid and carbohydrate metabolism. Among many other causes, the discrepancies between the studies may be related to the obesity status of the patients, their ethnicity, and the severity of the disease. Hence, to reliably identify microbial metabolites as potential diagnostic biomarkers in NAFLD, studies should be conducted repeatedly in cohorts with the same characteristics and with a larger number of patients.

One of the most important factors driving the metabolism of the gut microbes is the diet. Besides the influence of long-term dietary intakes, diet can also very rapidly change the microbiome [77]. It is also well known that hypercaloric diets promote the onset of NAFLD [78]. Therefore, it is surprising that besides our study [47], none of the other studies reviewed here compared dietary intakes between the study groups. Thus, it cannot be known whether the reported metabolic differences in the gut microbiota between the study groups are solely or partly due to dietary differences. In the future, diet should definitely be considered in studies presenting results from fecal metagenomics and metabolomics so that the microbiome could be included in the possible future diagnostics of NAFLD. Interestingly, it seems that in undernourished NAFLD patients, altered bile acid signatures are consistently reported [79]. Bauer et al. effectively proposed that collaborative, multi-omics approaches could improve hepatic health in an undernourished population. Henceforth, similar approaches should also be considered for over nourished Western populations suffering from NAFLD.

## Figures and Tables

**Figure 1 ijms-24-04855-f001:**
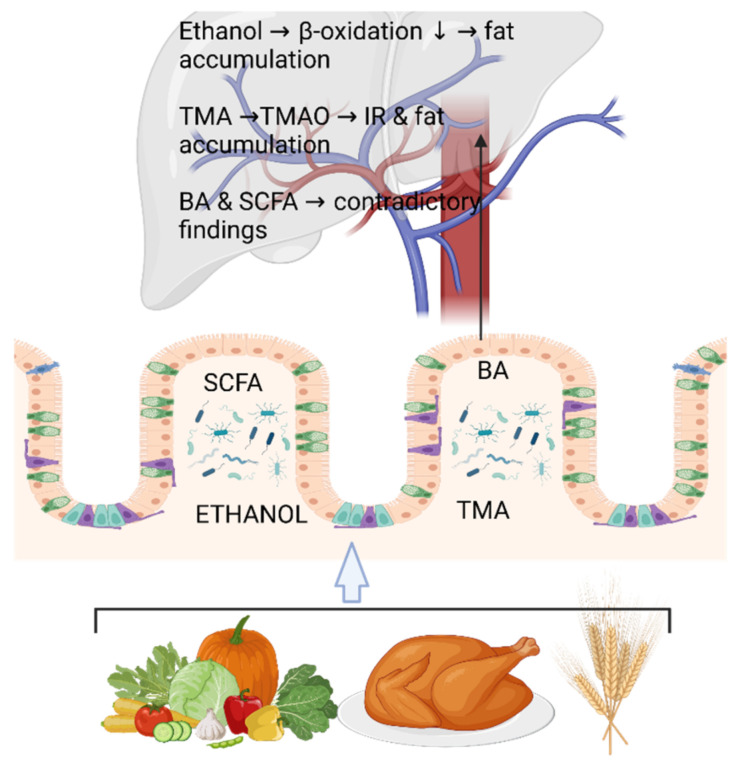
The abundance of many members of the gut microbiota affects the human physiology by processing the ingested food into certain bioactive metabolites. These molecules can act as inter-tissue signaling messengers by penetrating the portal vein and subsequently, the liver, to promote or prevent hepatic fat accumulation. For instance, dietary fiber is metabolized by the gut microbiota into short-chain fatty acids (SCFA, mainly butyrate, acetate, and propionate), and ethanol is produced from dietary carbohydrates. Gut microbiota produce trimethyl amine (TMA) from dietary choline, which is mainly derived from meat, yolk, and dairy products. TMA is further converted into trimethyl amineoxide (TMAO) in the liver. Primary bile acids (BA) are synthesized in the liver, stored in the gallbladder, and then released into the gut, where they are converted into secondary BA by the gut microbiota. In the liver, ethanol can increase fat accumulation by reducing beta-oxidation. TMAO has been shown to induce insulin resistance (IR) and further fat accumulation in the liver. Controversial findings on the role of BA and SCFA in hepatic fat accumulation exist. These will be reviewed in this article. The figure was created using BioRender.

**Figure 2 ijms-24-04855-f002:**
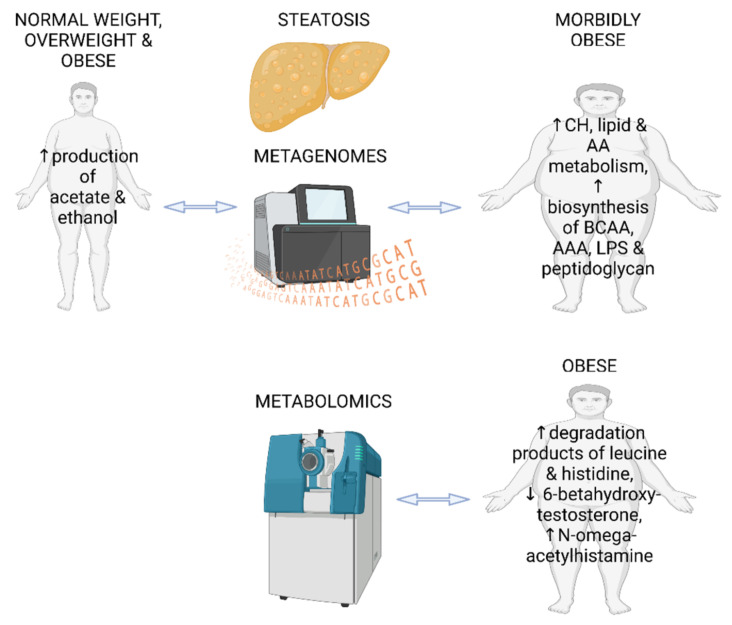
Main findings from two fecal metagenomic studies and one metabolomics study in individuals with hepatic steatosis and without diagnosed NAFLD. CH—carbohydrates; AA—amino acids; BCAA—branched chain amino acids; AAA—aromatic amino acids; LPS—lipopolysaccharides. The figure was created using BioRender.

**Figure 3 ijms-24-04855-f003:**
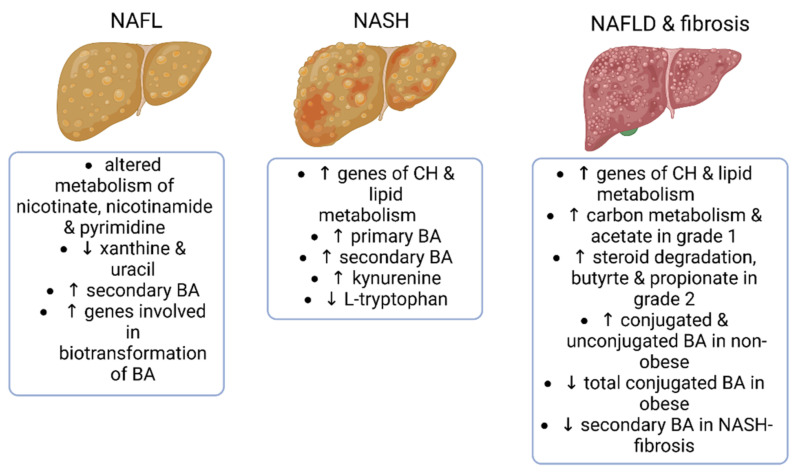
Main findings from studies reporting fecal metagenomics and/or metabolomics in NAFLD and advanced fibrosis. NAFL—non-alcoholic fatty liver; NASH—non-alcoholic steatohepatosis; CH—carbohydrates; BA—bile acids. The figure was created using BioRender.

## Data Availability

Not applicable.

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
