# Peer review of "Fecal Metagenomics and Metabolomics Identifying Microbial Signatures in Non-Alcoholic Fatty Liver Disease"

_ijms, 2023, doi:10.3390/ijms24054855_

Round 1
Reviewer 1 Report
The aim of this review article is to summarize the existing literature regarding the role of microbiota, via test of fecal samples using omics techniques, on the detection and development of non-alcoholic fatty liver disease (NAFLD).
The author has made a comprehensive review of the literature, including nice and illustrative figures and his/her expert opinion.
I would only suggest to add a Table with the studies mentioned on the review article, so that the reader could easily go and check the references on the Table.
Author Response
I wish to thank both reviewers for carefully going through the manuscript, and for the nice comments on it. Below you can find my answers to the specific comments, and the explanation of what has been changed, how and where. In addition to this, minor changes have been made during the spell check as suggested by both reviewers. All changes made to the manuscript are shown with “track changes”.
REVIEWER 1
The aim of this review article is to summarize the existing literature regarding the role of microbiota, via test of fecal samples using omics techniques, on the detection and development of non-alcoholic fatty liver disease (NAFLD).
The author has made a comprehensive review of the literature, including nice and illustrative figures and his/her expert opinion.
I would only suggest to add a Table with the studies mentioned on the review article, so that the reader could easily go and check the references on the Table.
My answer: I sincerely thank the reviewer for the nice comments on this manuscript. Thank you for the suggestion, I have added a table as requested. I placed it after the section “4.1. Fecal metabolomics and metagenomics identifying the gut microbial signatures in steatotic adults without diagnosed NAFLD”. However, if the editors see that it could be placed better to the end of the manuscript, it can be moved there.
Reviewer 2 Report
Overall, this is a well written review publication and offers a good overview of microbial signatures in NAFLD. However, before publication some points need to be clarified.
My comments:
Line 41 – Please change “histologically” to “histopathologically” .
Line 57 – Plaease add short methodology of this review.
Line 80 – why some letters in “hypertension”, “ALT” etc. are written bold?
Line 104 - From histological point of view there are only four kinds of tissues: epithelial, connective, muscular and nervous. Therefore, such term as “adipose tissue” (title, line 59) is not justified.
Line 152 – T5KO was already abbreviated in line 139
Line 174 – the names of bacteraias phyla shoud be written in italics.
Line 184, 188 – please explain what TLR4 and TLR2 stand for.
Line 248 – please provide more details about PICRUSt.
Line 317 – why names of these chemicals are written in italics?
Author Response
I wish to thank both reviewers for carefully going through the manuscript, and for the nice comments on it. Below you can find my answers to the specific comments, and the explanation of what has been changed, how and where. In addition to this, minor changes have been made during the spell check as suggested by both reviewers. All changes made to the manuscript are shown with “track changes”.
REVIEWER 2
Overall, this is a well written review publication and offers a good overview of microbial signatures in NAFLD. However, before publication some points need to be clarified.
My answer: I thank the reviewer for the nice comment on the manuscript.
My comments:
Line 41 – Please change “histologically” to “histopathologically” .
My answer: Thank you for pointing out the error. The word has been changed correctly in line 41.
Line 57 – Plaease add short methodology of this review.
My answer: Thank you for the suggestion. A short methodology has been added to the lines 59-63 as follows:
The literature searches for this review article were made between September and December 2022. The search words “fecal AND metabolomics AND (liver fat OR NAFLD OR NASH)” and “fecal AND metagenomics AND (liver fat OR NAFLD OR NASH)” were used both in PubMed and Ovid Medline. Studies in animals were omitted from the search results.
Line 80 – why some letters in “hypertension”, “ALT” etc. are written bold?
My answer: The intention was to highlight from which letters does the abbreviation come. However, for clarity all bolded letters have been modified.
Line 104 - From histological point of view there are only four kinds of tissues: epithelial, connective, muscular and nervous. Therefore, such term as “adipose tissue” (title, line 59) is not justified.
My answer: Thank you for the good comment. Adipose tissue has been replaced by connective tissue in line 107.
Line 152 – T5KO was already abbreviated in line 139
My answer: This has been corrected in line 157.
Line 174 – the names of bacteraias phyla shoud be written in italics.
My answer: The author believes that all taxonomic names in the text are now written in italics.
Line 184, 188 – please explain what TLR4 and TLR2 stand for.
My answer: I apologize for the missing information. The information has now been added in lines 196 and 200, respectively.
Line 248 – please provide more details about PICRUSt.
My answer: Thank you for the suggestion. In lines 260-264 the following information has been added:
The name is an abbreviation for Phylogenetic Investigation of Communities by Reconstruction of Unobserved States. PICRUSt is a bioinformatics software package designed to predict metagenome functional content from marker gene (e.g., 16S rRNA gene) surveys and full genomes using a developed algorithm developed by Languille et al.
Line 317 – why names of these chemicals are written in italics?
My answer: Thank you for pointing out the error. They are enzymes, and I have now corrected the names so that they are not in italics and added the corresponding bacterial gene names in italics. I hope this is clearer now.
